# The Role of Somatic Mutations on the Immune Response of the Tumor Microenvironment in Prostate Cancer

**DOI:** 10.3390/ijms22179550

**Published:** 2021-09-02

**Authors:** Camila Morais Melo, Thiago Vidotto, Luiz Paulo Chaves, William Lautert-Dutra, Rodolfo Borges dos Reis, Jeremy Andrew Squire

**Affiliations:** 1Department of Genetics, Medicine School of Ribeirão Preto, University of São Paulo, Ribeirão Preto 14048-900, SP, Brazil; camila.morais.melo@gmail.com (C.M.M.); thiagovidotto@gmail.com (T.V.); luizpaulocds@usp.br (L.P.C.); williamlautert@usp.br (W.L.-D.); 2Division of Urology, Department of Surgery and Anatomy, Medicine School of Ribeirão Preto, University of São Paulo, Ribeirão Preto 14048-900, SP, Brazil; rodolforeis@fmrp.usp.br; 3Department of Pathology and Molecular Medicine, Queen’s University, Kingston, ON K7L3N6, Canada

**Keywords:** immunotherapy, checkpoint blockade, immune evasion, innate and adaptive immune system, oncogenes, tumor suppressor genes, single-cell transcriptomics, mouse models of cancer, genomic instability, spatial imaging

## Abstract

Immunotherapy has improved patient survival in many types of cancer, but for prostate cancer, initial results with immunotherapy have been disappointing. Prostate cancer is considered an immunologically excluded or cold tumor, unable to generate an effective T-cell response against cancer cells. However, a small but significant percentage of patients do respond to immunotherapy, suggesting that some specific molecular subtypes of this tumor may have a better response to checkpoint inhibitors. Recent findings suggest that, in addition to their function as cancer genes, somatic mutations of *PTEN*, *TP53*, *RB1*, *CDK12*, and DNA repair, or specific activation of regulatory pathways, such as ETS or MYC, may also facilitate immune evasion of the host response against cancer. This review presents an update of recent discoveries about the role that the common somatic mutations can play in changing the tumor microenvironment and immune response against prostate cancer. We describe how detailed molecular genetic analyses of the tumor microenvironment of prostate cancer using mouse models and human tumors are providing new insights into the cell types and pathways mediating immune responses. These analyses are helping researchers to design drug combinations that are more likely to target the molecular and immunological pathways that underlie treatment failure.

## 1. Introduction

Previous approaches to cancer therapy were based mostly on how to control clonal proliferation of aggressive tumor cells. Recently, translational researchers have focused on the role of the anti-cancer immune system in the design of new therapies to improve the response and overall survival. Thus, an important new hallmark of cancer is understanding how to stop tumors bypass the host’s protective immune destruction mechanisms [1].

Immunotherapy acts by boosting the natural propensity of the immune system to recognize and respond to the presence of tumors [2]. Immunotherapy involves the use of specific checkpoint agents that block the suppressive interactions between a developing tumor and the defensive immune system of the patient [3]. The clinical responses to checkpoint blockade have been very successful across many different cancers [4]. Tumor features that have been associated with a good clinical response to anti-PD-1/PD-L1 checkpoint blockade therapies include T-cell infiltration, PD-L1 expression in tumor and/or immune cells, increased tumor mutational burden (TMB), and interferon gamma (IFNγ)-derived T-cell gene expression profiles [5]. However, the overall proportion of patients responding to immunotherapy remains modest, with single-agent response rates across different tumor types ranging from 10–35% [6]. In advanced prostate cancer (PCa), the effect of anti-PD-1 checkpoint blockade has been disappointing, with just 5–10% of tumors that exhibit defects in microsatellite instability (MSI) pathways [7] or *CDK12* biallelic mutations responding to this therapy [8].

Recent evidence suggests that factors influencing the success of checkpoint blockade therapy includes the absence of a pre-existing tumor-specific T cell response, or the exclusion of T-cells from the tumor microenvironment. To predict whether a particular cancer type will respond to immunotherapy, cellular classification systems have been developed in recent years based on the type of immune-cell infiltrate, density, and location within the tumor. Computational analyses of transcriptomic data have also been used to classify the immune content of tumors. As described below, these classifications are starting to provide some clues about the mechanisms tumors are using to avoid the host immune system. In Section 7, we discuss how these research advances are being applied in recent immunotherapy trials.

Key somatic mutations in PCa include fusions of *TMPRSS2* with *ETS* family genes, amplification of the *MYC* oncogene, deletion and/or mutation of *PTEN*, *RB1*, and *TP53* in advanced disease, together with amplification and/or mutation of the androgen receptor (AR) [9]. The immune system is able recognize and destroy tumor cells that are constantly arising during the life of an individual using a finely tuned anti-cancer immunosurveillance system. An emerging hypothesis to explain why most PCa fail to respond to immunotherapy is that specific somatic mutations acquired during tumor development induce changes in surrounding non-tumor cells of the microenvironment that help the tumor to evade immunosurveillance, the mechanism by which immune cells interact [10]. Recent reviews addressing this area in PCa have concentrated on advances in the use of checkpoint inhibitors [11], the roles of various immune subsets [12], DNA repair, mutational burden and tumor response [13], and the role of different cellular contexts on the immune microenvironment [14,15]. We will review the impact of the commonly acquired somatic mutations on immune response and the TME of PCa that are currently being used to identify potential predictive molecular biomarkers for immunotherapy clinical trials.

## 2. Interactions between the Tumor Microenvironment and Cancer

Tumor cells stimulate significant molecular, cellular, and physical changes in their surrounding microenvironment. This means that the cellular composition of the tumor microenvironment (TME) continuously evolves and can become more complex as tumors develop. In addition, there is variation in the cellular content of the TME of different tumor types, but a consistent hallmark of the TME of cancers includes immune cells, stromal cells, blood vessels, and an extracellular matrix [1]. New information about the TME indicates that it should now be considered “an active promoter of cancer progression” rather than earlier views of it being a passive supporting structure [16].

Early in tumor growth, a dynamic and reciprocal relationship develops between cancer cells and the TME. These interactions influence immune responses against cancer, local invasion, and subsequent metastatic dissemination. Tumors become infiltrated with diverse adaptive and innate immune cells that can perform both pro- and anti-tumorigenic functions. In addition, their activities are pivotal in mediating or suppressing host anti-tumor immune responses in the TME.

For cancers that are resistant to immunotherapy, recent discoveries suggest that tumors adopt several different strategies to avoid anticancer immune responses. For example, analyses of pre-treatment melanoma biopsies have demonstrated that checkpoint blockade response [17] correlated with the presence of tumor-infiltrating CD8+ lymphocytes at the invasive tumor margins. In head and neck cancer, the presence of an interferon-gamma (IFNγ) gene expression signature correlated with a better clinical response to immunotherapy [18]. Expression of IFNγ responsive genes is considered to represent an ‘inflamed’ transcriptional state in the TME, since these genes are associated with chemokine expression, antigen presentation, and cytotoxic effector molecules. These observations led to a TME classification of tumor-immune phenotypes involving three major classes of tumors: ‘immune deserts’ tumors (or ‘cold tumors’), showing no immune cell infiltration; ‘immune-excluded’ tumors (also cold) with immune cells at the tumor margins; and ‘inflamed’ tumors (or ‘hot’ tumors), showing immune infiltrates in the tumor core that can have IFNγ expression signatures [19,20,21] (Figure 1). Immune inflamed tumors, such as melanoma, are characterized by the infiltration of CD4+ and CD8+ T cells within the tumor parenchyma, suggesting the presence of a pre-existing anti-tumor response that has been dampened by an immunosuppressive microenvironment or by intrinsic T-cell tolerance. In contrast, cold tumors are characterized by the absence of pre-existing TILs and can be further subdivided into immune-excluded tumors, in which T cells have been attracted to the periphery of the tumor, but fail to infiltrate, and finally the immune-desert tumors, which have no detectable T-cell infiltrate [6].

Immunologically hot tumors, such as melanomas, are the most responsive to checkpoint blockade [17], and have a variety of infiltrating T cells in the TME and tumor [22]. In contrast, immunologically cold tumors have a low mutation load, lower and/or lowest probability of response to anti-PD1/PD-L1 treatment, are immune tolerant against self-antigens, and lack the T-cell-inflamed TME. PCa tumors are known to be sparsely infiltrated with T cells [23], suggesting that the poor immunotherapy response in PCa may be linked to their TME having an immune desert or excluded phenotype. When TILs are present in PCa, the presence of CD8+ lymphocytes have been reported to be a favorable prognostic indication [24], although an earlier study noted that the TILs present in PCa were ‘unresponsive’ or terminally differentiated [25]. A larger more recent study based on microarray gene expression profiles also reported the presence of high tumor TIL infiltrates were associated with worse distant metastasis-free survival [26].

To account for the contradictory findings in the PCa literature concerning lymphocyte density, it has been suggested that variation in the activation state of tumor specific T cells in the TME may account for the weak associations between TILs and the outcome [15]. Previous studies of PCa suggest that CD8+ lymphocytes in the TME may be inactive, suppressed, or unable to generate a functional cytotoxic response despite the presence of tumor antigen stimulation [27,28]. When looking at T-cell populations in PCa, studies have noted the presence of CD8+ CD25+ Treg cell clones that expressed FoxP3 and suppressed naive T-cell proliferation in prostate tumor-derived TILs, and there was a significant correlation between CD3+, CD8+, and FOXP3+ T-cell densities, but these were not associated with most clinical or pathologic variables. Increased T-cell density was significantly associated with *ERG* positivity and also with *PTEN* loss in the combined cohort of matched European-American and African-American ancestry patients [29,30].

Unfortunately, there is still no consensus concerning the best scoring methods amongst the large number of histologic studies of TILs. For example, some papers include only intratumoral TILs and others also score stromal TILs. For these reasons, there has been controversy about the prognostic value of scoring TILs in PCa, and there is limited appreciation of the functional status of lymphocytes in the TME [15]. It is possible that lymphocytes present in the TME are unable to perform immune activities due to mechanisms such as anergy (tolerance), exhaustion, or senescence [31]. As described below, more refined functional and transcriptomic classifications of effector cells in the microenvironment of human and murine PCa are being used to characterize the role that the TME plays in immune evasion and tumor progression.

## 3. Recent Lessons Learned about the Immune TME from Mouse Models of Human PCa

The development of an appropriate in vivo immunotherapy model systems for studying the cellular interactions between tumor and immune cells in the TME has been challenging because human tumors are usually grown as xenografts in immunocompromised mice. There are several recent investigations using genetically modified mouse models with functionally intact immune systems that are providing new information about underlying mechanisms of antitumor immune responses [32].

A comparison of murine model tumors to various human cancers has drawn attention to the role of immunomodulatory myeloid cells in the TME. These myeloid cells are essential for suppressing adaptive immunity and play a central role in ensuring that a developing malignancy is able to evade the host immune system [33]. Suppressive myeloid cells function by either direct cell–cell interaction with the target cells, such as T cells and NK cells, or through secreted factors. Two key classes of suppressive myeloid cells are the tumor-associated macrophages (TAMs) and myeloid-derived suppressor cells (MDSCs) [34].

The genetic background of tumors in mice promotes infiltration of specific immune-cell subsets into the TME of PCa [35]. Specifically, these authors showed that the double knockout of *Pml*, *Zbtb7a*, and *Pten* elevated expression of the neutrophil cytokine Cxcl5, leading to increased recruitment of these granulocytes into the TME. In contrast, *Tp53* knockout mice increased expression of another cytokine, C-X-C chemokine ligand 17 (Cxcl17), which led to recruitment of MDSCs. The expression data from murine PCa was similar to human tumors in the TCGA dataset. Moreover, the TME or murine PCa also had cellular features consistent with the ‘immune desert’ phenotype with very limited intratumoral immune infiltration.

MDSCs are emerging as key players in the immunosuppression of the TME of solid tumors. The main characteristic that defines MDSCs is their ability to inhibit immune responses, including those mediated by T cells, B cells, and natural killer (NK) cells [36]. Calcinotto et al. investigated whether MDSCs might contribute directly to resistance to anti-androgen therapy, using human PCa cells and mouse models of prostate cancer. They showed that high levels of the inflammatory cytokine Il-23 mediated paracrine effects on tumor-infiltrating MDSCs by activating downstream androgen receptor (AR) target genes through the signal transducer and activator of transcription 3 (STAT3)–RORγ signaling axis in tumor cells [37].

Tumor-associated macrophages (TAMs) comprise the macrophage populations surrounding and infiltrating solid tumors that are also closely involved in mediating immune responses against cancer [33]. TAMs can be polarized by various microenvironmental stimuli to generate a heterogeneous population with different properties and functions in the immune response [38]. Human cancer cells can influence TAMs polarization by releasing cytokines, glucocorticoids, extracellular vesicles, and extracellular matrix components that give rise to a large spectrum of pro-tumoral macrophages [39]. Dimitri et al. recently investigated the interplay between different macrophage activation states and tumor cells in the TME in *Pten* knockout PCa [40]. The TME of these tumors were highly infiltrated by TAMs that expressed the Type2 C-X-C chemokine receptor (Cxcr2), which led to an M2 anti-inflammatory polarized macrophages when activated with its ligand Cxcl2. They showed that these PCa tumors were sensitive to inhibition of Cxcr2, which induced TAMs to re-program towards an M1 anti-tumorigenic phenotype. The authors further highlighted the therapeutic potential for modulating TAMs in the TME by combining treatment with a Cxcr2 inhibitor with infusions of Cxcr2-KO activated monocytes, which further increased the efficiency of tumor inhibition [40].

There is an increasing need to understand the specific differences that may exist in human immune responses within metastatic sites and other tumor niches. Jiao et al. [41] developed a PCa mouse model to study the mechanisms that may underlie the development of different T-cell lineages. They found that tumors growing in bones contain high levels of IL-6 and TGFβ, which provide signals to change intratumoral CD4+ T cell lineage differentiation towards polarized to a Th17 cell subset rather than the Th1 lineage. Inhibition of TGFβ plus immune checkpoint blockade led to Th1 CD4+ T cell responses and the clonal expansion of CD8+ T cells in bone tumors, which improved anti-tumor responses and increased survival in mice. These data indicate that cytokines and growth factor levels within the TME may underlie different T-cell immune activities and variation in clinical outcomes to immunotherapy. In Section 5, we discuss factors leading to bone metastasis based on single-cell analysis of TAMs, which shows how specific cytokine cues in the TME promote growth of metastatic bone PCa [42].

Distinct subsets of tumor-associated myeloid cells may play a pro-tumorigenic role either by directly mediating chemotherapy response or by suppressing recognition of the adaptive immune system and facilitating immunotherapy resistance. The diverse roles that the various myeloid cell populations play in the TME is still poorly understood cells [36], but clues coming from advanced genomic studies described in the following sections. Current areas of active investigation are to determine what combinations of the common somatic mutations of PCa and altered downstream signaling pathways influence effector cells, such as MDSCs, TAMs, and cytokines, to create conditions in the TME that favor immune evasion.

## 4. Role of Somatic Mutations in Shaping the Microenvironment of PCa

The TCGA group used integrative multi-omics to identify six distinct immune subtypes across 33 cancers that were associated with prognosis, genetic, and immune modulatory alterations considered to shape the specific types of immune environments in the TME [43]. The genomics of the various tumor types studied expressed vastly different immune responses suggesting that tumor-specific mutational changes may strongly influence pathways of evasion and the tumor-immune interactions. Accumulating evidence suggests that some of the common somatic mutations of PCa not only impacts classical hallmarks of cancer pathways [1], but also elicits changes in the TME, allowing cancer cells to avoid immunosurveillance [16,44,45].

Prostate cancer has a high rate of genomic instability (amplifications, deletions, and chromosomal rearrangements) [46] but a relatively low mutational burden compared with other cancers. Only four driver genes (*ERG* fusion, *PTEN*, *TP53*, and *SPOP*) are recurrently mutated in more than 10% of primary tumors [9,13]. During the early stages, PCa is predominantly an androgen-dependent disease. Eventually, cancer cells adapt to castrate androgen levels to form recurrent disease mediated through *AR* reactivation, acquisition of additional alterations such as *PTEN* loss and PI3K activation (See Table 1).

### 4.1. Phosphatase and Tensin Homologue (PTEN)

The *PTEN* tumor suppressor gene regulates the *PI3K* pathway. *PTEN* deletion is present in 20–40% of PCa and loss is strongly associated with PCa progression [64] and immune response in cancer [15,56]. Our data in human PCa [55] suggest that PCa tumors with *PTEN* loss have changes in the TME that could facilitate immune evasion, with a higher density of FoxP3+ Tregs and more IDO1 protein expression in the stromal and tumor compartments in PTEN-deficient tumors compared to tumors that maintained PTEN activity. Studies of *Pten* in murine PCa demonstrated that its loss promotes negative regulation of expression of the immunosuppressive cytokines interleukin (IL)-10, IL-6, and vascular endothelial growth factor (VEGF) by inhibiting signal transducer and activator of transcription (STAT3). The *Pten*-null senescent PCa tumors from mice had high infiltration of MDSCs in the TME promoted by activation of the Janus kinase (JAK)2/STAT3 pathway and secretion of chemoattractant molecules to the TME [57,58]. A recent population-based study of >6000 PCa related *PTEN* loss to the statin pathway and inflammation and immune activation in lethal PCa [66]. These findings are supported by a new study of *Pten* knockout mice, which used methylation and transcriptomic profiling to show that loss of *PTEN* drives global changes in DNA CpG methylation with associated gene expression changes, thus affecting several inflammatory and immune molecular pathways during PCa development [65].

Gene fusions of the *ETS* transcription factor family (usually involving the *ERG* gene) are caused by genomic rearrangements with *TMPRSS2* gene and are found in over 50% of patients with clinically localized PCa [53]. As a result, the androgen-responsive promoter elements of *TMPRSS2* drive the expression of the *ETS* family transcription factors to promote PCa progression, invasion, and tumor aggressiveness, and lead to a significantly reduced survival [13,80,81,82]. In the context of *PTEN* loss, *ERG* can restore AR target gene expression and increases AR binding to target promoters, independent of AR protein levels and circulating levels of testosterone [83]. *Pten* knockout mouse experiments showed that *Ets* overexpression led to disease progression, as well as inducing a pro-inflammatory gene signature, both of which were impacted by *Ar* expression levels [84]. This finding suggests that the *Tmprss2-Erg* fusion promotes recruitment of regulatory T cells to the tumor site [50]. Recent studies have shown an association between low levels of TILs comparing *TMPRSS2*-*ERG* fusions producing chimeric transcripts in the coding in the fusion-positive cases results in suppression of immune response by alteration of cytokine production [51,52].

### 4.2. TP53

*TP53* mutations are found in 20–30% of PCa [61]. Loss of the TP53 protein has been shown to impact immunosuppression through increased PD1 and PDL-1 expression, and by activation of *NF-kB* signaling [60,85]. In primary PCa, the *TP53* missense mutation was associated with higher T-cell density [35,62] and related to increase genomic instability [63]. *Tp53* loss promotes the recruitment of tumor-supporting myeloid cells [35]. *Pten* loss in PCa models in combination with *Tp53* loss increases the secretion of Cxcl17, leading to the recruitment of MDSCs [35].

Sequencing data from TCGA database revealed mutations or homozygous deletions for DNA repair genes *MLH1, MSH2, BRCA2, BRCA1, ATM, CDK12, FANCD2,* and *RAD51B/C* in primary and metastatic prostate cancer [70]. The recognition that germline or somatic mutations in DNA repair genes occur in about 25% of patients with recurrent or advanced PCa [86] allows for the use of poly(ADP-ribose) polymerase (PARP) inhibitors in this PCa molecular subtype, since these drugs can selectively kill tumors with DNA repair defects [87]. Since RNA repair defects also affects tumor immunogenicity, there is emerging interest in using checkpoint blockades combined with PARP inhibitors, as described below.

### 4.3. CDK12

*CDK12* has been known to regulate DNA damage response genes (DDR) and modulate the expression of immune checkpoints, such as PD-1, in PCa. Mutations in *CDK12* are found in 1.2 to 1.5% and 4 to 6.9% of primary-PCa and mCRPC, respectively [8,67]. Wu et al. described a biallelic loss-of-function mutation in *CDK12* as a distinct molecular subtype of advanced PCa [8]. The genomic of PCa with *CDK12* mutations show a widespread pattern of acquired focal tandem duplication (FTD), scattered throughout the genome and enriched in gene-dense regions. The FTD in the genome yields tumor cells with enriched gene fusion, and fusion-induced neoantigen was presented at higher levels than observed in HR or ATM-prostate cancers. This molecular subtype also demonstrates a high level of immune cell infiltration and T-cell expansion than all other PCa. Rescigno et al. founded that CD4+ infiltrate higher in *CDK12* mCRPC group with a phenotype composed majority by CD4+ FOXP3-, which was associated with worse outcomes and immunosuppression [68]. Compared to CD4+ FOXP3-, the authors founded a low CD8+ infiltration load even with the high mutation burden in the *CDK12* mCRPC group.

### 4.4. AIRE

The immune gene encoding autoimmune regulator (*AIRE*) protein which confers autoimmune protection was regulated by androgen/AR complex, androgen recruits AR to the *AIRE* promoter and thereby enhances its transcription [88]. Interesting study reports show the *AIRE* differential expression in the androgen-sensitive cell line compared to androgen-insensitive cell line in addition to immune cell polarization [89]. In mouse models, *Aire* induced Il-6 and Pge2 switches the monocyte polarization to M2 phenotype [89].

Next, generation sequencing has also been pivotal in understanding the complexity and the extent of genomic diversity that occurs during PCa progression [90]. There is accumulating evidence that specific somatic mutations in PCa trigger downstream expression changes and the abundance of non-tumor cells and immune components in the TME, allowing cancer cells to avoid immunosurveillance (see Figure 2). More detailed understanding of how the various somatic mutations can influence the dynamic interplay between tumor cells and the diverse non-cancerous cells in the TME of PCa is being provided by sensitive new single-cell analytical technologies.

## 5. Single-Cell Descriptions of the Immune Ecosystem of the TME of Prostate Cancer

The TME of cancer has been compared to an ecosystem that involves the delicate interplay of various cell types, including tumor cells, immune cells, stromal cells, and many others [101]. The introduction of various powerful single-cell methodologies, such as cytometry with time-of-flight (CyTOF) analysis [102,103] and single-cell genomics [104], has provided unprecedented insights into the molecular biology of the dynamic cell interactions that characterize the TME of PCa.

Single-cell methods are typically performed for many hundreds to thousands of cells in a single experiment. Recent studies have shown that analyzing gene expression using single-cell RNA-seq (scRNA-seq) at the level of individual cells provides a much greater depth of analysis than earlier bulk methods [104]. When combined with DNA sequencing, scRNA-seq allows appreciation of the in vivo impact of genomic alterations on gene expression. Importantly, scRNA-seq can assess the mutational heterogenicity and transcriptional pathways in cell populations present in the bulk tumor and the TME in an unbiased fashion at the level of individual cells. There are a number of recent examples of scRNA-seq analysis of PCa, which have investigated the heterogeneity of tumors at the single-cell level [42,105,106].

In a study by Chen et al., scRNA-seq [42] was used to study both PCa tumor intracellular heterogeneity and to examine the tumor-infiltrating immune cell content of the TME. Some intriguing observations were made regarding the TME when they compared primary PCa to matched lymph node metastases, and to adjacent non-cancerous prostate tissue. They identified a subset of TAMs with high osteoclast (OC)-activity in all samples. Since OC mediates osteodegradation and bone is the preferred site for metastatic PCa [107], this finding provides new ideas for the role of TAMs in the promotion of metastatic PCa. As discussed above, TAMs are closely involved in mediating immune responses against cancer and can produce growth factors favoring tumor outgrowth [33]. The observations of this study imply that a subset of TAMs may promote metastatic progression and immune suppression at bony sites. These findings also agree with the mouse study of Jiao et al. [41] (discussed in Section 3) in which single-cell analyses were used to demonstrate lineage programming of T cells in favor of a bone microenvironment. Chen et al. also identified PSA (*KLK3*) expression within the antitumor effector CD8+ T cells in lymph node metastasis sites. This second observation was puzzling, as T cells were thought to be AR negative and *KLK3* is known to be an AR-dependent gene. They found that AR activity was absent in these T cells, and suggested that *KLK3* expression in effector CD8+ T cells was likely transferred from exogenous sources. In keeping with this hypothesis, they found enrichment of extracellular vesicles and exosome trafficking pathways, exclusively in the PSA positive T cells. This conclusion was in keeping with the emerging role of extracellular vesicles in exogenous modulation of immune responses in the TME [108].

Song et al. (2020) used scRNA-seq to compare *TMPRSS2-ERG* fusion positive tumors to ERG negative PCa [109]. They found common transcriptional pathways that were upregulated in the tumor, stroma, and CD4 T cell populations of *ERG* negative patient tumors, including the PD-1 and IFNγ signaling pathway, suggesting that *ERG* tumor cells may give rise to a distinct immune cell niche in the TME.

Vickman et al. [110] used scRNA-seq analysis to define the cellular heterogeneity in the stromal carcinoma-associated fibroblasts (CAF) from primary human PCa. They identified six unique subpopulations of CAFs with altered expression of various cytokines and a large CAF subpopulation that had elevated expression of *CCL2* and *CXCL12*. Their findings implicate a role for CAF and specific cytokine secretions in the TME for triggering inflammatory myeloid cell recruitment.

Another recent example of the power of single-cell analysis was the recent investigation of the collaborative impact of two different somatic mutations on the TME of PCa [111]. The tumor-suppressor gene *CHD1* encodes a chromatin remodeler, and inactivating mutations occurs in 7–10% of human PCa. Zhao et al. used a PCa mouse model to show that *Chd1* deletion resulted in less aggressive tumor growth and increased overall survival compared to prostate tumors in which *Pten* alone was deleted [111]. Single-cell CyTOF was used to provide high-dimensional analysis of immune cells and other TME components [102]. These data showed that PCa tumors with codeletion of *Pten* and *Chd1* had decreased levels of immunosuppressive MDSC and TAMs and increased levels of T, B, and NK cells. These single-cell observations were clarified by functional in vitro assays that indicated that *CHD1* may regulate MDSC recruitment by IL6. Their analysis showed that *CHD1* binds directly to the Il6 promoter at a conserved *CHD1*-binding motif, so that the promoter becomes more accessible for transcriptional activation and increased expression leading to recruitment of MDSC into the TME. These observations were consistent with the regulatory role of *CHD1* in establishing open chromatin. Importantly, their findings suggest that inhibiting the CHD1–IL6 signaling axis may improve PCa responses to immune checkpoint blockade by reducing the presence of immune suppressive MDSCs in tumors.

A major limitation of all single-cell multi-omics analytical methods is that they require cell suspensions as input materials. This means that spatial information of the location of the cell within the tissue context of tumor or TME is lost during dissociation. New sequence-based spatial applications of single-cell analyses are addressing these deficiencies and providing new types of maps of the TME.

## 6. Spatial Analysis of the Immune Architecture of the TME of Prostate Cancer

Defining previously uncharacterized subsets of immune cells by single-cell analysis is crucial to the understanding of the expression and functions of cells within the ecosystem of the TME. As mentioned previously, cell interactions and spatial relationships cannot be investigated directly by these approaches since tissue architecture and cell-to-cell contact is destroyed by pre-processing for suspension-based cell assays, such as scRNA-seq and CyTOF. Several of the studies referenced above showed that the heterogeneity and different cellular activities of the myeloid subtypes and other immune cell phenotypes in the TME were partly based on their physical orientation within the tumor or in the surrounding the periphery of the tumor [40,41,112,113]. These findings underscore the need for more accurate descriptions of the effects of somatic mutations on anti-tumoral immunity based on descriptions of the spatial organization of the various components of the immune system in the TME of PCa.

New imaging technologies with high dimensionality and greater computational resolving power have recently been combined with the detection power of single-cell sequencing to provide detailed molecular maps of the TME [114]. The research potential of these combined technologies was immediately recognized leading to Spatially resolved transcriptomics being named ‘Method of the year’ in 2020 [115]. There are several recent excellent reviews that describe different aspect of this technology [116,117], and the various platforms available [118].

The spatial architecture of immune cells in the TME can now be described in greater detail because of these advanced imaging technologies. Spatial maps of the TME contain information on the distances between cellular neighborhood within the tumor section and immune cells of interest (for example, MDSCs or specific TAM subtypes). These methods can also pinpoint and count all the TILs in relation to specific gross morphologic features or ‘cellular neighborhoods’ within the TME. Examples of PCa neighborhoods of interest for spatial mapping might be areas of focal high Gleason score, regions with perineural invasion, or margins with capsular tumor growth. For example, regions with different Gleason scores were recently shown to have distinct gene expression signatures that appear to be related to local capacity for cellular proliferation and invasion in higher Gleason score cells [119]. In addition, more detailed spatial imaging of tumor cell types of interest such as neuroendocrine, basal, or luminal cells can be directly related to the activity states of lymphocytes and other effector cells in the vicinity. Any distance-dependent metric of immune interactions such as paracrine, autocrine, or inter-cell contacts can be measured by high-resolution spatial approaches.

The first spatial transcriptomics maps of PCa used imaging methods that approached single-cell resolution [120]. They mapped gene expression in sections of primary PCa and detected activation of signaling pathways related to stress, inflammation, and angiogenesis at the tumor–TME interface periphery. A more recent spatial study focused on intratumoral gene expression and differences in cell phenotype both across and within different metastatic PCa [121]. Their analysis reports a general absence of immune cell infiltrates and a lack of PD1, PD-L1, and CTLA4 expression in the majority of metastases. There was high expression of the immune checkpoint proteins B7-H3/CD276 in areas of high AR expression.

Calagua et al. [122] studied >100 aggressive primary PCa with multiplex spatial immunofluorescence and found that 25% of tumors expressed PD-L1 and contained a high density of TILs. These lymphocytes contained subpopulations of exhausted progenitor CD8+ T cells and differentiated effector T cells. A subset of these cells was found to express TCF1, indicative of exhausted progenitor T cells that may be more likely to respond to immunotherapy. Significantly, these cells were found close to antigen-presenting cell niches within the tumor sections. Genomic analysis showed significant enrichment for somatic mutations of *RB1* and *BRCA2* and deep deletions in *CHD1*. These results suggest that a subset of localized PCa have genomic features indicating they are more likely to be immunogenic and respond to checkpoint blockade drugs.

Keam et al. [123] used multiplex spatial analysis of sections of localized PCa to map out the effects of high-dose brachytherapy. They found that many immune checkpoint molecules (e.g., B7-H3, CTLA4, PDL1, and PDL2) and TGFβ levels were increased in response to radiation. They then used a published 16-gene tumor inflammation signature [18] to differentiate tumors into their intrinsic immune activation states. Their analysis showed that most localized PCa were phenotypically cold tumors before brachytherapy. Significantly, after radiation exposure, 80% of tumors had converted into more immunologically activated ‘hot’ tumor tissue, with accompanying spatially organized immune infiltrates in the sections that were accompanied by consistent signaling changes. Their findings that ‘cold’ PCa can be converted into ‘hot’ tumors has important implications for future clinical trials involving high dose radiation and immunotherapy.

## 7. Recent Immunotherapy Clinical Trials in Prostate Cancer

Until recently, PCa immunotherapy has shown only modest efficacy for patients. However, now, with the increased understanding of immune mechanisms operating in the TME, immunotherapy in PCa is viewed more positively, especially for CRPC. Recent reviews addressing this area in PCa have concentrated on advances using immunotherapy combined with other approaches that facilitate the action of checkpoint blockade drugs [3,9,11].

The clinical trials phase I and II with programmed death receptor 1 (PD-1) and cytotoxic T-lymphocyte antigen-4 (CTLA-4) inhibitors have shown limited benefits [124,125,126]. The most promising strategies for mCRPC treatment combining of two different checkpoint inhibitors or combining one checkpoint inhibitor with enzalutamide, an anti-androgen approved for use in mCRPC [124,125,126]. Recent studies investigating the combination of anti-PD-1 and anti-CTLA4 had promising results in phase I/II and III [127,128,129] with the autologous cellular immunotherapy sipuleucel-T [130]. This treatment consists of autologous peripheral blood mononuclear cells, including antigen-presenting cells, which are previously activated with a recombinant fusion protein that is a prostate antigen fused to granulocyte–macrophage colony-stimulating factor, an immune-cell activator [131,132].

There are six therapies approved by the U.S. Food and Drug Administration (FDA) that have been shown to improve overall survival in men with mCRPC: docetaxel, sipuleucel-T, abiraterone, enzalutamide, cabazitaxel, and radium-223 [125] (Table 2).

There is currently considerable interest surrounding whether the combination of PD-1 and CTLA-4 inhibitors would be more efficacious in these genomically selected patients with *CDK12* mutation [8,133]. Wu et al. showed that two out of four patients who received PD-L1 blockers had decreased PSA level and shrinkage of metastases [8]. In a retrospective multi-center study, Antonarakis et al. sought to clinically characterize *CDK12* loss of function in PCa treated with a PD-1 inhibitor [133]. They found that *CDK12*-altered prostate is an aggressive subtype with poor outcomes to first-line therapy. In the same study, nine men received a PD-1 inhibitor (pembrolizumab, [*n* = 5], nivolumab [*n* = 4]); 33% of mCRPC treated patients had a PSA response and median of progression-free survival of 5.4 months [134]. In a retrospective study, Schweizer et al. described the differences between standard treatments and ICB in *CDK12* mCRPC [134]. In this study, nineteen mCRPC patients received ICB (pembrolizumab [*n* = 15], atezolizumab [*n* = 1], ipilimumab + nivolumab [*n* = 1], tremelimumab + durvalumab [*n* = 1] and atezolizumab [*n* = 1]); 11/19 (59%) had a decline in PSA level, with 2 patients (11%) having 100% PSA decline. Therefore, tumors harboring *CDK12* loss of function seem to be more aggressive and invasive than other PCa subtypes and responded to ICB. However, there were some patients from this subgroup who were non-responsive to ICB treatment despite harboring *CDK12* alterations.

Another class of genetic alteration that has drawn attention for combinatorial use of immunotherapy are the mutations in the DNA repair pathway genes. Combinations of PARP inhibitors in tumors with mutations in homologous recombination genes may result in loss of compensatory DNA repair mechanisms, leading to death of cancer cell due to unrepairable DNA damage [135,136].

There is recent interest in new classes of immunotherapeutic drugs that take advantage of interactions of the innate immune system between cancer cells and macrophages through the CD47 and the signal regulatory protein alpha protein (SIRPα) [137,138]. CD47 is a molecule expressed by nearly all normal tissues and serves as a marker of self-recognition. When bound to SIRPα, located on the surface of macrophages, CD47 triggers anti-phagocytic signals. New immunotherapy drugs, such as ALX148 [139], comprise a SIRPα fusion protein that binds to CD47, which enhances phagocytosis of tumor cells by macrophages. These types of drugs are just starting to be applied in advanced solid tumors such as CRPC.

## 8. Conclusions

Improved understanding of the tumor biology of immune response has been greatly facilitated by recent technological advances in the analysis of the PCa microenvironment. Detailed transcriptome characterization of tumor cells and the non-cancerous cells in the TME is providing important clues about the role of somatic mutations may play in modulating the crosstalk between cancer cells and the immune system. Studies of PCa using single-cell sequencing, spatial immunobiomarkers, and functional analyses in model systems are collectively providing novel insights into the composition, function, and location of immune cells and other non-cancerous cells within the TME. These ongoing studies in mouse models and human tumors are designed to investigate the best predictive somatic mutation biomarkers for immunotherapy in advanced PCa. Molecular profiling is increasingly guiding treatment decisions and sequential treatments of advanced cancers. The application of new immunobiomarker assays in PCa promises continuing improvements in immunotherapy success in future years.

## Figures and Tables

**Figure 1 ijms-22-09550-f001:**
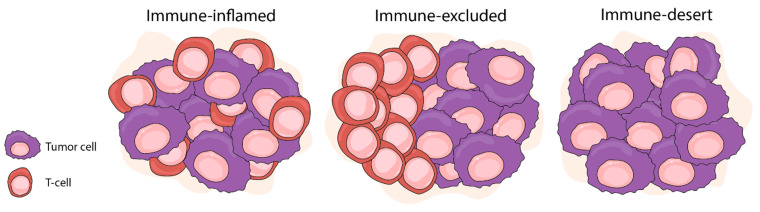
Immunological classification of tumors—inflamed (left) with high degree of cytotoxic T-cell infiltration, immune-excluded (center) with presence of T cells at invasive margins but absence in tumor tissues, and immune desert (right) absence of T cells within tumor and at margins. Prostate cancer has features of both an immune desert and an excluded phenotype (figure based on ANANDAPPA; WU; OTT, 2020) [6].

**Figure 2 ijms-22-09550-f002:**
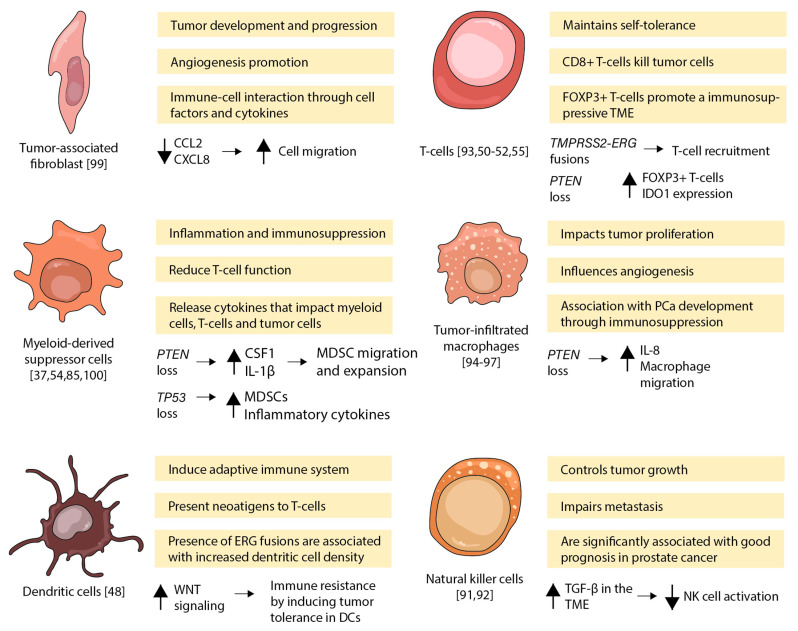
Mutation and gene expression changes leading to cellular and immune evasion effects in the TME of PCa. The various somatic mutations can influence the non-cancerous cell in the tumor microenvironment of prostate cancer. CCL2 (C-C Motif Chemokine Ligand 2), CD8+ T cell (cytotoxic T lymphocyte), CSF1 (colony Stimulating Factor 1), CXCL8 (C-X-C Motif Chemokine Ligand 8), DC (dendritic cells), ERG (ETS Transcription Factor ERG), FoxP3+ (Forkhead box p3), FoxP3+ T cell (regulatory T lymphocyte), IDO1 (indoleamine 2,3-Dioxygenase 1), IL-1β (interleukin 1, beta), IL-8 (interleukin 8), MDSC (myeloid-derived suppressor cell), NK (natural killer cell), PTEN (phosphatase and tensin homologue), TGFβ (transforming growth factor beta), TME (tumor microenvironment), TMPRSS2 (Transmembrane Serine Protease 2), TP53 (Protein Coding), WNT (important pathway for immune cell maintenance and renewal). Up arrows (increase). Down arrows (decrease) [37,48,50,51,52,54,55,85,91,92,93,94,95,96,97,98,99,100].

**Table 1 ijms-22-09550-t001:** Somatic mutations and immune response in PCa microenvironment.

Somatic Mutations	Frequency of PCa Mutated	Proposed Responses of Immune Effectors in PCa Microenvironment	References
**Genes**			
*ETS* fusion genes	20–30%	Activation *NOTCH*, *WNT* and *NF-kB* signaling; Induction of a pro-inflammatory gene signature (Tregs and dendritic cells recruitment).	[47,48,49,50,51,52,53]
*PTEN* loss	20–40%	Increased PDL1 and IDO1, FoxP3+ Tregs, resting dendritic cells, M1 macrophages, MDSCs, and CAFs; PI3K pathway regulation; Negative regulation of cytokines.	[15,54,55,56,57,58,59]
*TP53* loss	20–30%	Induction of a pro-inflammatory and immunosuppressive TME (TAM, MDSCs, T cell); Increased PD1 and PDL-1 expression; Activation of NF-kB signaling; Increased genomic instability.	[35,55,60,61,62,63,64,65,66]
*CDK12* inactivation	1.2–6.9%	Secretion of pro-tumorigenic cytokines (CCL18, /IL-8); Immune checkpoints modulation; Increased neoantigen load and immune infiltration (T cell (higher CD4+ and lower CD8+), TAM and MDSCs).	[8,67,68]
*SPOP*/*CHD1*	6–15%	Upregulation of the AR, PI3K/mTOR, ERG fusion and PD-L1; Correlated to deletion of CHD1.	[13,69]
*AR* gene mutation	1%	Increased immune infiltration (Tregs and IL-10) and lower CD3+ and CD68+ cells; Decreased in IFNγ-secretion by T cells; IL-23 secreted by MDSCs can cause downstream activation of AR target genes.	[70,71,72]
*MYC* amplification	20%	Pro-inflammatory and immunosuppressive TME (recruitment of TAMs and MDSCs); Induce genomic instability.	[37,73,74]
*RB1* loss	28–30%	Activation of tumor cell cycle and antigen presentation; IFN response; Reduce the immune cells mobilization and NK cells.	[63,75,76]
**Pathways**			
PI3K activation	20–40%	Pro-inflammatory and immunosuppressive TME (recruitment of TAMs, MDSCs and PD-L1).	[37,77]
DNA repair	19%	Accumulation of genetic aberrations, tumor evolution and progression; Inflamed TME phenotype; Increase IFN pathways and interactions between tumor-specific antigens and immune cells.	[78]
WNT/b-catenin activation	20%	Immunosuppressive TME (recruitment of FOXP3+ regulatory T cell and lower CD8+:FoxP3+ ratio).	[79]

**Table 2 ijms-22-09550-t002:** Ongoing clinical trials (Phase III) in prostate cancer.

Agent	Function—Target	Description	NCT Number
Sipuleucel-T	Personalized, autologous, and cellular immunotherapy. Target the prostate specific antigen (PSA).	To determine if sipuleucel-T was effective for PCa treatment. Patients with localized PCa, metastatic castration sensitive PCa or mCRPC.	NCT00779402 NCT00065442 NCT03686683 NCT01133704 NCT00005947
Ipilimumab	Humanized IgG1 monoclonal antibody that blocks cytotoxic T lymphocyte antigen-4 (CTLA-4) and removes an inhibitory signal from reducing the activity of T lymphocytes.	To evaluate whether the addition of radiotherapy or chemotherapy plus ADT improves the prognosis and survival of patients. Patients with localized PCa, metastatic castration sensitive PCa or mCRPC.	NCT00861614 NCT03879122 NCT01057810
Aglatimagene Besadenovec + Valacyclovir	ProstAtak^®^ kills tumor cells and stimulates a cancer vaccine effect. Killing tumor cells in an immune stimulatory environment induces the body’s immune system to detect and destroy cancer cells.	To evaluate the effectiveness of ProstAtak^®^ immunotherapy in combination with radiation therapy for patients with intermediate-high risk localized prostate cancer.	NCT01436968
Bevacizumab + Cisplatin + Gemcitabine + Hydrocloride	Humanized monoclonal IgG antibody and inhibits angiogenesis by binding and neutralizing VEGF-A.	To evaluate the effectiveness of bevacizumab to induce changes in body’s immune system and to interfere with the ability of tumor cells to grow and spread. Patients with metastatic castration sensitive PCa.	NCT00942331
PROST-VAC-V + PROST-VAC-F + GM-CSF	Active immunotherapy vaccine that contains PSA as the tumor-associated antigen used to generate a T-cell response against prostate cancer.	To determine whether PROSTVAC alone or in combination with GM-CSF is effective in prolonging overall survival in men with few or no symptoms from metastatic, castrate-resistant prostate cancer.	NCT01322490
DCVAC + Docetaxel + Taxotere	Cell therapy platform designed to improve efficacy compared to earlier generations of dendritic cell therapies by targeting multiple antigens and applying an immune-stimulatory technique.	To confirm the hypothesis that the combination of docetaxel with DCVAC/PCa followed by a maintenance therapy with DCVAC/PCa would improve overall survival in patients with metastatic castration-resistant prostate cancer.	NCT02111577

## Data Availability

Not applicable.

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
