# Peer review of "The Role of Somatic Mutations on the Immune Response of the Tumor Microenvironment in Prostate Cancer"

_ijms, 2021, doi:10.3390/ijms22179550_

Round 1

Reviewer 1 Report

In this review, the authors described the immune microenvironment of prostate cancer and the role of somatic mutations on the regulation of tumor microenvironment. The authors also showed recent findings through novel technologies such as single cell analysis and spatial analysis. The tables in the review were helpful in summarizing the role of somatic mutations. The only issue is that there are too many words in the tables. So it might be better to have some figures to help demonstrate the interaction between the tumor microenvironment and cancer cells.

Author Response

Point 1: In this review, the authors described the immune microenvironment of prostate cancer and the role of somatic mutations on the regulation of tumor microenvironment. The authors also showed recent findings through novel technologies such as single cell analysis and spatial analysis. The tables in the review were helpful in summarizing the role of somatic mutations. The only issue is that there are too many words in the tables. So, it might be better to have some figures to help demonstrate the interaction between the tumor microenvironment and cancer cells.

Response 1: We agree that the tables contained too many words and were not concisely organized. So, we reorganized tables 1 and 3, and decreased the number of words inside the tables keeping the coherence and important information. Table 2 has been replaced by a new Figure 2, in which we illustrate how the various somatic mutations can influence the non-cancerous cell in the tumor microenvironment of prostate cancer. Figure 1 has been modified slightly in appearance to have the same style as the new Figure 2.

Reviewer 2 Report

I read with great interest the manuscript by Camila M. Melo et al. It is timely and well written review article. The Authors tunely described the role of somatic mutation tumour on the immune response of the tumour microenvironment in prostate cancer.

It could be interesting discuss whether differences have been seen or reported on the relationship between pathological Gleason score and the tumour microenvironment. The Authors introduced the Gleason score in a sentence (pag. 11 433-435)

Only minor grammatical errors have been found throughout the manuscript.

Author Response

Point 1: I read with great interest the manuscript by Camila M. Melo et al. It is timely and well written review article. The Authors tunely described the role of somatic mutation tumour on the immune response of the tumour microenvironment in prostate cancer. It could be interesting discuss whether differences have been seen or reported on the relationship between pathological Gleason score and the tumour microenvironment. The Authors introduced the Gleason score in a sentence (pag. 11 433-435).

Response 1: We briefly covered the pathological Gleason score and the relationship with the tumor microenvironment. However, the reviewer 2 made an important point and we now provide more information on the subject on the page 10, 444-447 as suggested. See below:

“Examples of PCa neighborhoods of interest for spatial mapping might be areas of focal high Gleason score, regions with perineural invasion, or margins with capsular tumor growth. For example, regions with different Gleason scores were recently shown to have distinct gene expression signatures that appear to be related to local capacity for cellular proliferation and invasion in higher Gleason score cells (Georgescu 2016). In addition, more detailed spatial imaging of tumor cell types of interest such as neuroendocrine, basal or luminal cells can be directly related to the activity states of lymphocytes and other effector cells in the vicinity.”

Georgescu, I.; Gooding, R.J.; Doiron, R.C.; Day, A.; Selvarajah, S.; Davidson, C.; Berman, D.M.; Park, P.C. Molecular characterization of Gleason patterns 3 and 4 prostate cancer using reverse Warburg effect-associated genes. Cancer Metab. 2016, 4, 1–12, doi:10.1186/s40170-016-0149-5.

Point 2: Only minor grammatical errors have been found throughout the manuscript.

Response 2: We revise the manuscript and correct the grammatical errors throughout the text (highlighted in yellow).

Reviewer 3 Report

The author adequately and timely described about recent progress regarding understanding of immunological microenvironment of prostate cancers and its therapeutic application particularly T cells and cancer cell interaction.  However, recently much attention to the interaction of cancer cells and innate immune cells such as  macrophages, particularly through Cd47/Sirpalpha axis (Trends Immunol. 2018 Mar;39(3):173-184; Semin Oncol. 2020 Apr-Jun;47(2-3):117-124).  It would be better to describe some points above related to prostate cancers and innate immune cell interaction and its potential therapeutic application.

Author Response

Point 1: The author adequately and timely described about recent progress regarding understanding of immunological microenvironment of prostate cancers and its therapeutic application particularly T cells and cancer cell interaction. However, recently much attention to the interaction of cancer cells and innate immune cells such as macrophages, particularly through Cd47/Sirpalpha axis (Trends Immunol. 2018 Mar;39(3):173-184; Semin Oncol. 2020 Apr-Jun;47(2-3):117-124). It would be better to describe some points above related to prostate cancers and innate immune cell interaction and its potential therapeutic application.

Response 1: We thank reviewer 3 for bringing up this important new aspect of checkpoint inhibition therapies.  We provide a new paragraph at the end of section 7 (“Recent immunotherapy clinical trials in prostate cancer”) page 13, 527-535. See below:

“There is recent interest in new classes of immunotherapeutic drugs that take advantage of interactions of the innate immune system between cancer cells and macrophages through the CD47 and the signal regulatory protein alpha protein (SIRPα) (Logtenberg 2020; Oronsky 2020). CD47 is a molecule expressed by nearly all normal tissues and serves as a marker of self-recognition. When bound to SIRPα, located on the surface of macrophages, CD47 triggers anti-phagocytic signals. New immunotherapy drugs, such as ALX148 (Chow 2020) 20, 2020), comprise a SIRPα fusion protein that binds to CD47, which enhances phagocytosis of tumor cells by macrophages.  These types of drugs are just starting to be applied in advanced solid tumors such as CRPC.”

Logtenberg, M.E.W.; Scheeren, F.A.; Schumacher, T.N. The CD47-SIRP α immune checkpoint. Immunity 2020, 52, 742–752, doi:10.1016/j.immuni.2020.04.011.

Oronsky, B.; Carter, C.; Reid, T.; Brinkhaus, F.; Knox, S.J. Just eat it: A review of CD47 and SIRP-α antagonism. Semin. Oncol. 2020, 47, 117–124, doi:10.1053/j.seminoncol.2020.05.009.

Chow, L.Q.M.; Gainor, J.F.; Lakhani, N.J.; Lee, K.W.; Chung, H.C.; Lee, J.; LoRusso, P.; Bang, Y.-J.; Hodi, F.S.; Santana-Davila, R.; et al. A phase I study of ALX148, a CD47 blocker, in combination with standard anticancer antibodies and chemotherapy regimens in patients with advanced malignancy. J. Clin. Oncol. 2020, 38, 3056–3056, doi:10.1200/JCO.2020.38.15_suppl.3056 Journal of Clinical Oncology 38, no. 15_suppl (May 20, 2020) 3056-3056.